# A Hybrid Stainless-Steel SPME Microneedle Electrode Sensor for Dual Electrochemical and GC-MS Analysis

**DOI:** 10.3390/s23042317

**Published:** 2023-02-19

**Authors:** Samuel M. Mugo, Scott V. Robertson, Marika Wood

**Affiliations:** Department of Physical Sciences, MacEwan University, Edmonton, AB T5J 4S2, Canada

**Keywords:** solid phase microextraction (SPME), dual electrochemical and GC-MS detection, modified stainless steel microneedle (MN)

## Abstract

A mechanically robust in-tube stainless steel microneedle solid phase microextraction (SPME) platform for dual electrochemical and chromatographic detection has been demonstrated. The SPME microneedle was fabricated by layer-by-layer (LbL) in-tube coating, consisting of carbon nanotube (CNT)/cellulose nanocrystal (CNC) film layered with an electrically conductive polyaniline (PANI) hydrogel layer (PANI@CNT/CNC SPME microneedle (MN)). The PANI@CNT/CNC SPME MN showed effective analysis of caffeine by GC-MS with an LOD of 26 mg/L and excellent precision across the dynamic range. Additionally, the PANI@CNT/CNC SPME MN demonstrated a 67% increase in sensitivity compared to a commercial SPME fiber, while being highly robust for repeated use without loss in performance. For electrochemical detection, the PANI@CNT/CNC SPME MN showed excellent performance for the detection of 3-caffeoylquinic acid (3-CQA). The dynamic range and limits of detection (LOD) for 3-CQA analysis were 75–448 mg/L and 11 mg/L, respectively. The PANI@CNT/CNC SPME MN was demonstrated to accurately determine the caffeine content and 3-CQA in tea samples and dark roast coffee, respectively. The PANI@CNT/CNC SPME MN was used for semiquantitative antioxidant determination and composition analysis in kiwi fruit using electrochemistry and SPME-coupled GC-MS, respectively.

## 1. Introduction

The development of effective and rapid techniques for preconcentration, isolation, and detection of target compounds is of great importance in the field of analytical chemistry [1]. The solid-phase microextraction (SPME) technique coupled with GC-MS fits this criterion. Functionally, SPME relies on the principles of sorption and desorption, where a target compound binds to an SPME apparatus (typically a polymeric sorptive fiber), which can be subsequently injected into a GC-MS or HPLC instrument for analysis [1,2]. SPME has a wide scope of analytical uses, including urine and blood analysis [3], pollutant monitoring [4], and food quality assurance [5]. A wide variety of sorptive substrate materials may be used for SPME fabrication, such as polydimethylsiloxane (PDMS) [1,6], carbon nanotubes (CNT) [7], conductive polymers [8,9], molecularly imprinted polymers (MIPs) [10], and metal-organic frameworks [11]. Different materials can be especially advantageous, for instance, CNT are mechanically durable and have a high surface area for enhanced compound sorption [2]. Conductive polymers, such as polypyrrole, have an affinity for the sorption of charged analytes, where desorption can be controlled by an applied voltage [12]. Additionally, the chemical structure of certain conductive polymers may lend well for sorption of polar and conjugated compounds alike [9].

Electrochemical sensors are advantageous for chemical analysis due to their rapid response times and portability. In contrast, chromatographic techniques are lab-localized and suffer from slow analysis times. Due to their electrocatalytic activity and conductivity, CNTs are favorable for use in electrochemical sensing [13]. CNT-based electrochemical sensors can measure a wide range of electroactive analytes, including, but not limited to, microbes [14], pesticides [15], hormones [16], and antioxidants [17]. Similarly, polyaniline (PANI), a conductive polymer, has several advantages, thus making it useful for incorporation into electrochemical sensing devices such as inherent electrical conductivity and stability [18]. Structurally, PANI can exist in several oxidation states, including oxidized (pernigraniline), partially oxidized (emeraldine and nigraniline), and reduced forms (leucoemeraldine) [19]. The scope of application of PANI-based electrodes is wide, with literature examples including detection of glucose [20], lactate/pH [21], hydrogen peroxide [22], and heavy metal [23]. Due to their applications in both SPME and electrochemical sensing, CNT- or PANI-based SPME fibers may allow for simultaneous electrochemical and SPME-coupled GC-MS, thereby extending their scope of analytical applications. This integrated approach provides a unique synergy with an electrochemical detection ideal for rapid chemical screening, while GC-MS is useful for qualitative and confirmative analytics. 

SPME remains a viable technique for diverse analytes extraction. Improvements in SPME tailorable materials chemistry is necessary to extend the scope of their applications. However, the high cost (~$250) per fiber [24] and fragility of commercial SPME fibers remains a significant disadvantage. In this work, we demonstrate a frugal and mechanically robust PANI@CNT/CNC SPME microneedle (MN) useful for electrochemical detection and for coupling to GC-MS, yielding a platform for dual complementary chemical analysis. 

The SPME stainless steel MN were fabricated via layer-by-layer (LbL) in-tube coating and were used in either electrochemical and/or SPME-coupled GC-MS analysis of different organic compounds in coffee, tea, and fruit samples. The use of the PANI@CNT/CNC SPME MN for dual electrochemical and SPME-coupled GC-MS was demonstrated using kiwi fruit, where in situ semi-quantitative antioxidant determination and organic composition analysis at different fruit sample penetration depths were performed. Figure 1 depicts a schematic for the dual SPME sensor platform for electrochemical and GC-MS detection.

## 2. Experimental

### 2.1. Materials

Carboxylic acid functionalized carbon nanotubes (CNT) (diameter 9.5 nm, length 1.5 μm), potassium ferricyanide (K₃[Fe(CN)₆]), potassium chloride (KCl), (3-glycidyloxypropyl)-trimethoxysilane (GOPS), 3-(trimethoxysilyl) propyl methacrylate (TPM), caffeine, 13C-isotopically labeled caffeine, 3-caffeoylquinic acid (3-CQA), and ammonium peroxydisulfate (APS), monobasic dihydrogen phosphate (NaH_2_PO_4_), dibasic monohydrogen phosphate (Na_2_HPO_4_), and glacial acetic acid were purchased from Sigma Aldrich, Markham, ON Canada. All the reagents purchased were of ACS grade or highest purity possible. The Supelco Divinylbenzene/Carboxen/Polydimethylsiloxane (DVB/CAR/PDMS) SPME fiber assembly (needle size 24 ga) (57348-U) was used as the commercial SPME fiber. Aniline was bought from Fisher Scientific, Hampton, NH, USA, and cellulose nanocrystals (CNC) were donated by Alberta Innovates, Calgary, AB, Canada. Stainless steel hypodermic needles (0.7 mm × 40 mm) were bought from a local pharmacy. All reagents were of analytical reagent grade. All aqueous solutions were prepared using Milli-Q Ultrapure de-ionized water (DI). The kiwi fruits were obtained from a local grocery store in Edmonton, AB, Canada. Dark roast coffee, green, and earl grey tea were purchased from a local coffee shop in Edmonton, AB, Canada.

### 2.2. Fabrication of the PANI@CNT/CNC SPME MN

First, the stainless-steel hypodermic MN were coated LbL using previously demonstrated protocol with minor modifications [25,26,27]. Briefly, the stainless-steel MN were immersed into piranha solution consisting of 1:1 H_2_SO_4_:H_2_O_2_ (*v*/*v*) for 4 h, which functionalized the metal surface with active hydroxyl moieties. The MNs were then thoroughly washed in DI water, and subsequently silylated by immersion for 4 h in an aqueous solution consisting of TPM, DI water, and methanol in a 2:1:8 (*v*/*v*/*v*) ratio. Silylation resulted in chemical binding of organosilane moieties to the MN surface. This was followed by a 1.5 mL infusion of a CNT/CNC (0.10%/0.40%) homogenous suspension (in DI water) using a syringe pump at a flow rate of 15 µL/min, which led to the formation of a thin CNT/CNC within the MNs. PANI conductive layer was them polymerized within the CNT/CNC coated MN by infusing (at 15 µL/min) 1.5 mL of 0.10 M aniline prepolymer mixture (in 1 M H_2_SO_4_) mixed with 50 µL of 0.10 M APS initiator, with the reaction carried out in an ice bath (4 °C). Lastly, to stabilize the conductive layers on the MN surface, 0.50 mL of a 2:3:5 GOPS:acetic acid:DI water solution was infused through the MN and allowed to dry overnight at room temperature, yielding the PANI@CNT/CNC SPME MN. 

### 2.3. Instrumentation

All the electrochemical measurements, including cyclic voltammetry (CV) and Electrochemical Impedance Spectroscopy (EIS), were carried out using a BASi Palmsens-4 potentiostat (PalmSens B.V., Houten, The Netherlands) in a three-electrode system with a platinum wire as the auxiliary electrode, and an in-house Ag/AgCl reference needle [25,26,27]. Scanning electron microscopy (SEM) of microneedle electrodes was carried out on Zeiss Sigma 300 VP field emission scanning electron micrograph. FTIR spectra were recorded on Bruker Tensor 27 FTIR instrument fitted with diamond attenuated total reflectance (ATR). Raman spectra were measured using Bruker SENTERRA Raman microscope. An Agilent 6890 GC with a HP-5ms column (30 m length, 0.25 mm internal diameter, 0.25 µm film thickness) coupled with a single quadrupole Agilent 5975C VL MSD (Agilent Technologies, Santa Clara, CA, USA) was used to perform the GC-MS analysis.

### 2.4. Electrochemical Characterization and Analysis

Electrochemical measurements were carried out in a three-electrode electrochemical cell comprised of an PANI@CNT/CNC SPME MN, an in-house Ag/AgCl reference needle, and a platinum wire as the working, reference, and counter electrodes, respectively. The electroactive surface area of the fabricated sensors was calculated by performing CV at a different scan rate from 25 mV/s to 500 mV/s in 25 mM K_3_[Fe(CN)_6_], and the data was modeled with the Randles–Sevcik equation [25,26,27]. Furthermore, all the microneedle sensors were characterized using EIS to obtain the corresponding electron transfer resistances (R_ct_). EIS was performed in 5 mM K_3_[Fe(CN)_6_] in the 20.0–200,000.0 Hz frequency range, with a sinusoidal amplitude of 6.0 mV. 

For analyte standards analysis, the electrochemical cell contained 10 mL of 0.10 M phosphate buffer (pH = 7.1). In a typical analytical procedure, 75 µL aliquots of 1.0 × 10^4^ mg/L of 3-CQA standard were successively spiked into the electrochemical cell, with four CVs being acquired after 1 min of equilibration. All CV measurements were performed in the −1.0 to 1.0 V range, at a scan rate of 0.10 V/s. The signal was determined by averaging the Faradaic current in the −0.50–0 V range, subtracting the blank current signal, and dividing this value by the blank current. To evaluate the PANI@CNT/CNC SPME MN electrode stability and reusability under electrochemical conditions, we further tested triplicate sensors over a one-month period, analyzing a 300 mg/L 3-CQA 15 times, about once every two days, and the sensor response data distribution was determined. Each standard run was analyzed in triplicate. 

### 2.5. PANI@CNT/CNC SPME MN Extraction Procedure and GC-MS Analysis

The PANI@CNT/CNC SPME MN was coupled with GC-MS for analysis of caffeine. In a typical procedure, the SPME MN was allowed to equilibrate in a caffeine standard for 10 min with light stirring to allow for sorption of the analyte. The caffeine standards analyzed were in the 97.6–400 mg/L concentration range and were spiked with 75 µL of 5.0 mg/mL isotopically labeled caffeine-(trimethyl-^13^C_3_) as an internal standard. The CNT/CNC SPME fiber was then directly injected into the GC-MS inlet port for 3 min for desorption to take place. A commercial SPME fiber was similarly utilized for comparison. 

For caffeine analysis using the PANI@CNT/CNC SPME MN, the GC-MS inlet port was set to a temperature of 290 °C with splitless injection mode. Helium was used as the carrier gas at a flow of 1.0 mL/min. The oven temperature was set to 85 °C initially and ramped up to 180 °C (30 °C/min) over 3.0 min, where it was then held until the end of the run (total run time = 13 min). The MS source and quad was set to 230 and 150 °C, respectively. The MS detector temperature was set at 200 °C. To evaluate the PANI@CNT/CNC SPME MN stability and reusability under the high temperature of the GC/MS conditions, a 400 mg/L caffeine standard spiked with internal standard was analyzed 15 times, about once every two days, and the SPME MN response data distribution was determined. Each caffeine standard run was analyzed in triplicate.

### 2.6. Kiwi, Coffee, and Tea Sample Analysis

The PANI@CNT/CNC SPME MN were also evaluated for analysis of fruit and coffee samples. Because of the high overpotential of caffeine, 3-CQA was the analyte of choice to evaluate effectiveness of the PANI@CNT/CNC SPME in detection in real sample environment. For electrochemical analysis of 3-CQA in coffee, the standard addition method was employed. First, 10 mL of 0.10 M phosphate buffer (pH 7.0) was analyzed using the SPME MN, and a blank CV was acquired. Then, 1.0 mL of dark roast coffee sample was introduced, followed by five 75 µL aliquots of a 1.0 × 10^4^ mg/L of 3-CQA standard, with the CV being acquired in quadruplicate after each new addition. The antioxidant content in kiwi fruit was electrochemically analyzed by poking the SPME MN working, reference, and auxiliary electrodes into the fruit and acquiring CV in quadruplicate (−0.9–0.9 V, 0.10 V/s).

For GC-MS analysis of caffeine in green and earl grey teas, the samples were spiked with isotopically labeled caffeine-(trimethyl-^13^C_3_) and analyzed similarly to the caffeine standards outlined above using a PANI@CNT/CNC SPME MN. Kiwi fruit was analyzed by probing either the kiwi surface or core for 10 min using a PANI@CNT/CNC SPME MN, and injecting it into GC-MS inlet port for a 3 min desorption. GC-MS settings for kiwi analysis were similar to those for the caffeine analysis, except that the oven was set at 50 °C initially and was ramped up to 200 °C at 10 °C/min.

## 3. Results & Discussion

### 3.1. Materials Characterization

The morphology of the successful LbL fabrication of the PANI@CNT/CNC SPME MN was verified by SEM. Figure 2a shows the tip of the CNT/CNC coated MN, where the rough surface of the CNT/CNC composite coating is visible. Magnification of the CNT/CNC coated MN reveals a physically entangled porous interwoven network structure between the CNTs and CNCs (Figure 2b), held together by strong hydrogen bonding interactions. The magnified SEM image of the PANI@CNT/CNC SPME MN is shown in Figure 2c, where the rough and porous morphology of the electrically conductive PANI matrix is evident.

To further verify the successful LbL assembly of the PANI@CNT/CNC SPME MN, Raman spectroscopy was performed following each stage of development. Two sharp peaks at 1350 and 1585 cm^−1^ were observed in both CNT/CNC and PANI@CNT/CNC SPME MNs (Figure 3), corresponding to the D- and G-bands of the CNTs, respectively. The D band is generally attributed to the disordered carbon structure of the CNTs, while the G band corresponds to its C-C bond stretching [25,26,28]. The Raman spectra for the PANI@CNT/CNC SPME MN exhibited a small, sharp peak around 1032 cm^−1^ (Figure 3), which was attributed to both the C-H bending of the benzenoid structure, and the C=N stretching in quinoid structure of PANI [29]. Moreover, the intensity of the D- and G-band peaks decreased after fabrication of PANI (Figure 3). These results confirm the successful assembly of the PANI@CNT/CNC SPME MN. These results from characterization of the PANI@CNT/CNC SPME MN were comparable to a similar needle platform involving PANI coating reported elsewhere [26]. This attests to the reproducibility in the fabrication of the conductive polymer coatings with the PANI@CNT/CNC SPME MN platform. 

### 3.2. Electrochemical Characterization

The electroactive surface area for each stage of the PANI@CNT/CNC SPME MNs development was evaluated by performing CV at different scan rates (0.025 to 0.20 V/s) in 5.0 mM K_3_FeCN_6_ (in 0.10 M KCl) and invoking the Randles–Sevcik equation [30]. Figure 4a,b show the difference in CV response, the relationship between the cathodic peak current (i_p_), and the square root of the scan rate for CNT/CNC and PANI@CNT/CNC SPME MNs, respectively. The calculated electroactive surface areas for the PANI@CNT/CNC and CNT/CNC SPME MNs are shown in Table 1. The increase in the electroactive surface area of the PANI@CNT/CNC SPME MN relative to the CNT/CNC coated MN (Table 1) is indicative of the enhanced conductivity and porosity afforded by the PANI layer. The determined electroactive surface area of 0.0215 cm^2^ was statistically identical to a similar PANI-based MN coated platform reported earlier for a different application, which confirms excellent reproducibility in the fabrication process [26].

The PANI@CNT/CNC SPME MN was then characterized by EIS during the various stages of its LbL assembly using 25 mM K_3_FeCN_6_ (in 0.10 M KCl) as a standard redox probe. Nyquist plots and equivalent circuit fittings for the silylated MN, CNT/CNC, and PANI@CNT/CNC SPME MNs are shown in Figure 5 and Appendix A, respectively. A low electron transfer resistance (R_ct_) value is desirable, as it is indicative of greater signal transducing capabilities and electrical conductivity [25]. R_ct_ values for the PANI@CNT/CNC SPME MNs, CNT/CNC coated MN, and silylated MNs were extracted from the circuit fittings (Appendix A) and are shown in Table 1. There was a decrease in the R_ct_ value after integration of CNT/CNC into the silylated MN, attributed to the increased electroactive surface area afforded by the conductive CNT/CNC composite film. The R_ct_ further decreased upon addition of PANI to the CNT/CNC coated MN. When composited with PANI, the CNT nanostructures act synergistically with PANI, forming conductive bridges between their emeraldine salt domains, and thus increasing the electrical conductivity of the PANI@CNT/CNC SPME MN [25,31]. These values bear similarities to those earlier-reported in a similar MN platform [26].

### 3.3. PANI@CNT/CNC SPME MN-Coupled GC-MS Analysis

The ability of the PANI@CNT/CNC SPME MN to adsorb caffeine was tested through coupling to GC-MS. For comparison, the ability of a commercial SPME fiber to adsorb caffeine was tested in tandem with the PANI@CNT/CNC SPME MN. Caffeine is an alkaloid compound commonly found in coffee and tea, and it acts as a stimulant with potential antioxidant effects [32]. Overlapped extracted ion chromatograms (EICs) for caffeine acquired using the PANI@CNT/CNC SPME MN and commercial SPME fibers are shown in Figure 6a,b, respectively. Excellent linear regressions for PANI@CNT/CNC SPME MN and commercial SPME, were observed (Figure 6c) across caffeine standard, 5–400 mg/L, with a% RSD ranging from 1.4% to 5.0% across this concentration range, when each standard was analyzed in triplicate, indicative of excellent precision.

The PANI@CNT/CNC SPME MN had improved caffeine adsorption capabilities, evidenced by a 67% increase in sensitivity compared to the commercial SPME fiber, as determined by the ratio of the calibration sensitivities between the two platforms (Figure 6c). The increased affinity for caffeine for the PANI@CNT/CNC SPME MN compared to the commercial SPME based on DVB/CAR/PDMS can be attributed to the inherent high surface area impacted by the synergy of PANI@CNT/CNC hybrid materials for the SPME MN platform. This is consistent with literature where some conductive polymers have found to have excellent sorption for polar analytes [2,9]. The LOD of the PANI@CNT/CNC SPME MN was 26 mg/L, which is well below the typical concentration of caffeine in coffee and teas. Figure 6d shows the mass spectra of extraction ion EIC for *m*/*z* 194 (caffeine standard) and *m*/*z* 197 (caffeine isotopic label internal standard) peaks. 

### 3.4. Electrochemical Analysis of 3-CQA

The CV response of the PANI@CNT/CNC SPME MN to 3-CQA was then examined. 3-CQA is one compound in the chlorogenic acid family with well-known antioxidant properties [32,33,34]. Overlapped CV responses of the PANI@CNT/CNC SPME MN to phosphate buffer blank, 75, and 450 mg/L 3-CQA are shown in Figure 7a. The peak current signal associated with 3-CQA oxidation was averaged in the −0.5–0.0 V and plotted as a function of concentration as shown in Figure 7b. The% RSD in the linear concentration ranged from 3.0% to 1.7%, when each standard was analyzed in triplicate, indicative of excellent precision. The PANI@CNT/CNC SPME MN responds to 3-CQA with a sensitivity of 0.0030 µA/mgL^−1^ (Figure 7b), and LOD of 11 mg/L, which is well below the typical concentrations in coffee and other fruit samples [32,33]. 

### 3.5. Electrochemical and PANI@CNT/CNC SPME MN-Coupled GC-MS Analysis of Kiwi, Tea, and Coffee Samples

The ability of PANI@CNT/CNC SPME MN to analyze fruit and beverage samples in situ using both electrochemistry and GC-MS was then examined. First, the PANI@CNT/CNC SPME MN was used to poke into a kiwi fruit, along with a reference and counter electrode as shown in Figure 8a, and were used to semi-quantitatively determine the combined antioxidant content at different penetration depths and ripening times of the kiwi using CV. Different CV responses between 0-day and 3-day kiwi fruit were observed as shown in Figure 8b, potentially due to the changes in antioxidant capacity due to fluctuation of compounds such as ascorbic acid that typically occurs as the fruit ripens [35,36]. Following CV analysis, the PANI@CNT/CNC SPME MN was coupled with GC-MS to identify various compounds found within kiwi fruit. Labeled total ion chromatograms (TICs) for GC-MS analysis of kiwi surface and core are shown in Figure 9a,b, respectively. Associated lists of potential compounds identified through NIST library database in each kiwi sample depth analysis are shown in Figure 9c,d. Volatile esters, such as the methyl and ethyl esters of butanoic acid [37], and various sugar esters were easily identified (Figure 9). This verifies the ability of the PANI@CNT/CNC SPME MN to be dually usable as an electrochemical sensor and for SPME GC/MS analysis. The fact that the PANI@CNT/CNC are entrained within the stainless steel needle safeguards the SPME electrode from destruction during sample penetration, extraction, and GC/MS injection, hence their evidenced robustness. 

Next, the PANI@CNT/CNC SPME MN was used to electrochemically determine the 3-CQA content in dark roast coffee. Figure 10a show the overlaid CVs obtained from the analysis of 0.10 M phosphate buffer blank, coffee (in 0.1 M phosphate buffer), and 330 mg/L 3-CQA (in 0.10 M phosphate buffer) analyzed using the PANI@CNT/CNC SPME MN. 3-CQA is one of the many antioxidant compounds contained within coffee, and its total content depends on the conditions of coffee preparation [32,33]. Figure 10b show associated standard addition plot of △Current as a function of 3-CQA concentration spiked in the beverage sample. The excellent linear regression (R^2^ of 0.99) of the calibration plot is evidence to the performance of PANI@CNT/CNC SPME MN in the analysis of 3-CQA in coffee, a real-world sample with complex matrices. The average% RSD across the different standard spikes in the coffee samples were 1.7%, indicative of excellent precision of the MN electrode. By using the standard addition calibration plot, the amount of 3-CQA was determined to be 2.2 ± 0.2 mg/mL, which aligns well with the total chlorogenic acid content found in regular dark roast coffee [33]. To confirm the reproducibility of the PANI@CNT/CNC MN sensors under electrochemical conditions, we further tested triplicate sensors for over one month period, analyzing a 300 mg/L 3-CQA for ~15 times, and the% RSD was determined to be ~1.7% for each day analysis and ~3.0% RSD, when data was averaged across all the runs for all the three sensor electrodes.

Using the PANI@CNT/CNC SPME MN sensor and the CV results demonstrated herewith, selectivity of electroactive analytes would be based on the peak currents at different voltages. While this does not yield perfect selectivity as certain electroactive analytes may have overlapping redox peaks, it should be noted that highly selective sensors can be obtained using the PANI@CNT/CNC MN platform by integrating additional layers of biological and biomimetic molecular receptors. For example, integration of molecularly imprinted polymers has been demonstrated using similar platforms by imprinting neonicotinoids, and thus the platform can be highly selective and versatile with additional analyte specific molecular receptor [27]. 

The PANI@CNT/CNC SPME MN was further coupled with GC-MS and used to determine the caffeine content in green tea and earl grey tea samples using the external calibration curves in Figure 6c. Overlaid EICs for both green and earl grey tea samples are shown in Appendix A. The caffeine content of green and earl grey tea samples were found to be 0.111 ± 0.007 and 0.132 ± 0.007 mg/mL, respectively. These values align well with the caffeine content of 1 min steeped green and earl grey teas determined by Chin et al., 2008 [38]. These results confirm the ability of PANI@CNT/CNC SPME MN for use in electrochemical sensing and GC-MS analysis of food and drink samples. To further evaluate the reproducibility of the PANI@CNT/CNC SPME MN under hot temperature GC/MS conditions, 400 mg/L of caffeine standard analyzed for 15 times over a period of one month using triplicate SPME MN platform, yielding an average of 2.5% RSD for triplicate platforms over this time, indicative of robustness even in high temperatures [25,26]. This is not surprising as CNT/CNC and PANI are known to have excellent thermal stability. The fact that the PANI@CNT/CNC is entrained within the stainless-steel needle, adds to the robustness.

## 4. Conclusions

A mechanically robust SPME MN comprising a hypodermic needle with a CNT/CNC and PANI in-tube coating for dual electrochemical and GC-MS analysis was demonstrated. The success LbL in-tube coating was confirmed using Raman and electrochemical techniques, with excellent reproducibility in the fabrication process. The PANI@CNT/CNC SPME MN demonstrated enhanced sensitivity towards the adsorption of caffeine compared to a commercial SPME fiber, while showing excellent robustness with repeated use in hot temperatures of the GC-MS conditions. Moreover, the PANI@CNT/CNC SPME MN was used to accurately determine the caffeine content of coffee and tea samples via coupling to GC-MS and could be evaluated in the future for other food and environmental applications. Electrochemically, the PANI@CNT/CNC SPME MN was used to quantify 3-CQA in coffee and to semi-quantitatively probe contents in a ripening kiwi fruit through dual electrochemical and GC-MS analysis. Future work will involve extending evaluating for other analytes in different real world environmental and food samples. To increase the selectivity for the PANI@CNT/CNC SPME MN dual platform to afford specificity to different analytes, biomimetic molecular receptors such as molecularly imprinted polymers (MIPs) can be easily integrated. Indeed, there are many literature studies on the molecular imprinted conductive polymers prepared from functional monomers such as an aniline and pyrrole for. As such, the frugal PANI@CNT/CNC SPME MN opens opportunities for dual chemical analytics through electrochemistry and GC-MS.

## Figures and Tables

**Figure 1 sensors-23-02317-f001:**
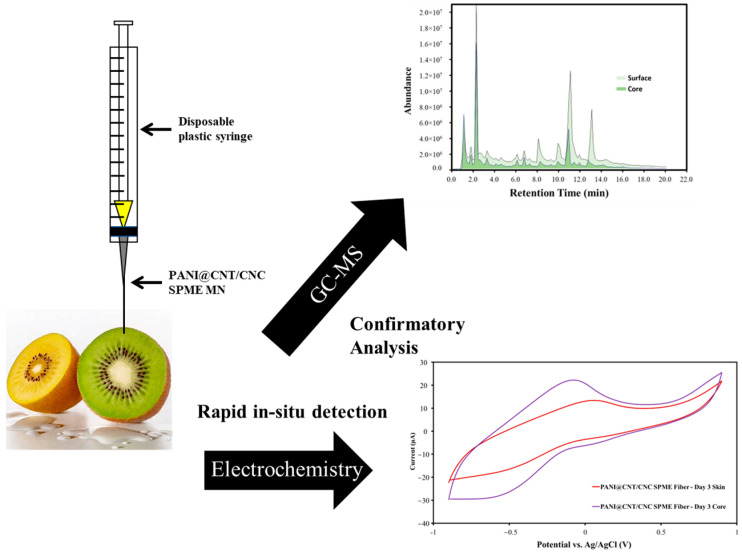
Schematic for the dual SPME sensor platform for electrochemical and GC-MS detection.

**Figure 2 sensors-23-02317-f002:**
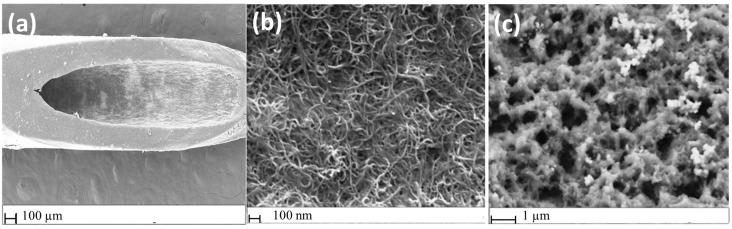
SEM images of (**a**) CNT/CNC coated MN; (**b**) magnified CNT/CNC coated MN; and (**c**) magnified PANI@CNT/CNC SPME MN.

**Figure 3 sensors-23-02317-f003:**
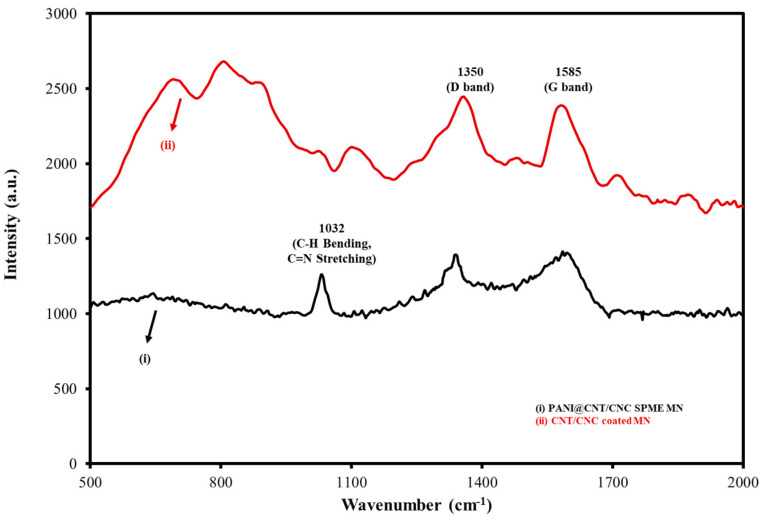
Overlaid Raman spectra for (i) PANI@CNT/CNC and (ii) CNT/CNC SPME MNs.

**Figure 4 sensors-23-02317-f004:**
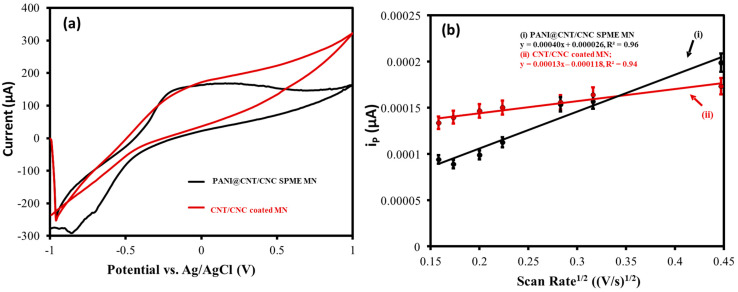
(**a**) Representative CV scans taken at 0.10 V/s and (**b**) Overlapped linear plot of peak cathodic current (i_p_) vs. square root of scan rate for PANI@CNT/CNC SPME MN and CNT/CNC coated MN acquired in 5.0 mM K_3_FeCN_6_ (in 0.1 M KCl).

**Figure 5 sensors-23-02317-f005:**
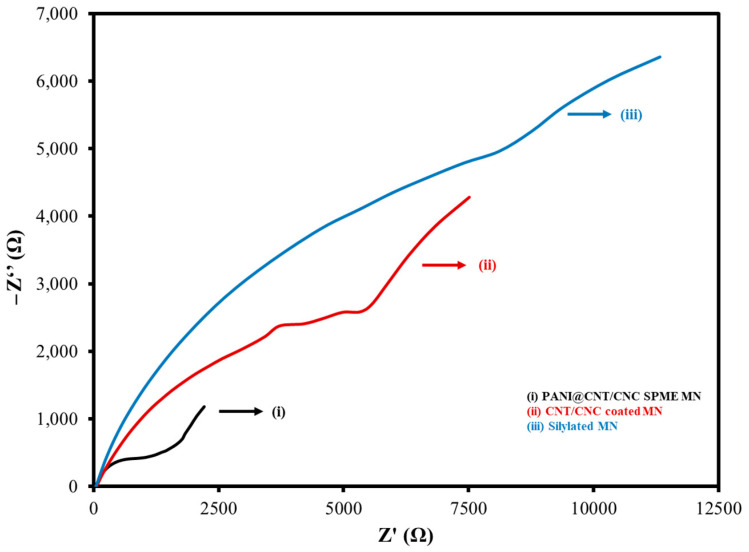
Overlaid Nyquist plots for (i) PANI@CNT/CNC SPME MN; (ii) CNT/CNC coated MN; and (iii) silylated MN taken in 25 mM K_3_FeCN_6_ (in 0.10 M KCl).

**Figure 6 sensors-23-02317-f006:**
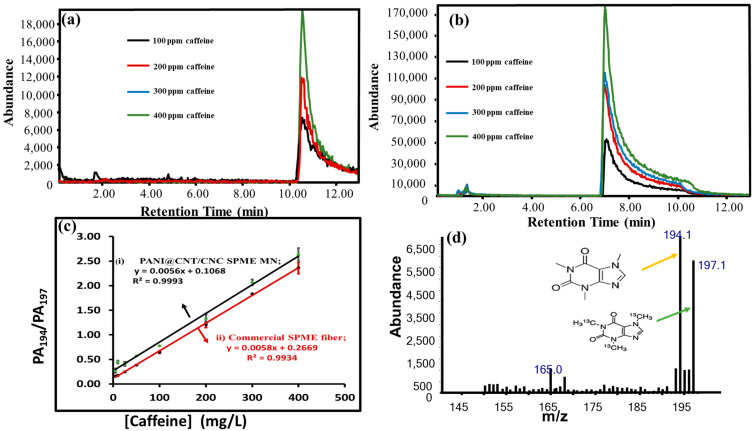
Overlaid extracted ion chromatograms (EIC) of GC-MS analysis of caffeine standards (*m*/*z* = 194 ± 0.30) acquired following SPME using the (**a**) PANI@CNT/CNC SPME MN and (**b**) commercial SPME; (**c**) associated caffeine concentration vs. peak area of caffeine-194/peak area of caffeine-197 (PA_194_/PA_197_) for caffeine extracted using the (i) PANI@CNT/CNC SPME MN and (ii) commercial SPME; (**d**) the mass spectra of extraction ion EIC for *m*/*z* 194 (caffeine standard) and *m*/*z* 197 (caffeine isotopic label internal standard) peaks.

**Figure 7 sensors-23-02317-f007:**
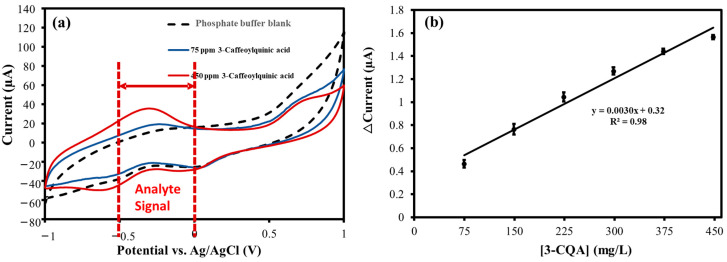
(**a**) Overlaid CVs obtained from the analysis of 0.10 M phosphate buffer blank, 75, and 450 mg/L 3-caffeoylquinic acid (3-CQA) (in 0.10 M phosphate buffer) taken using the PANI@CNT/CNC SPME MN. (**b**) Associated 3-CQA concentration vs. △Current linear plot determined by averaging the 3-CQA oxidation peak current signal in the −0.5 V–0.0 V.

**Figure 8 sensors-23-02317-f008:**
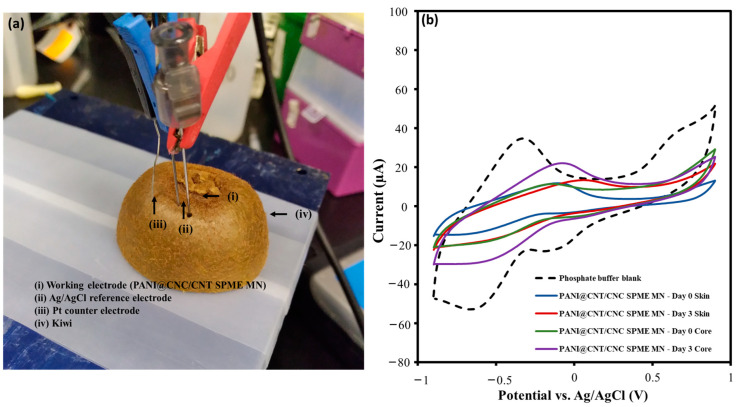
(**a**) Camera image of the experimental setup for in situ electrochemical kiwi analysis; (**b**) Overlaid CVs obtained from analysis of 0.10 M phosphate buffer blank, fresh (0-day), and 3-day ripened kiwi fruit skin and core using the PANI@CNT/CNC SPME MN.

**Figure 9 sensors-23-02317-f009:**
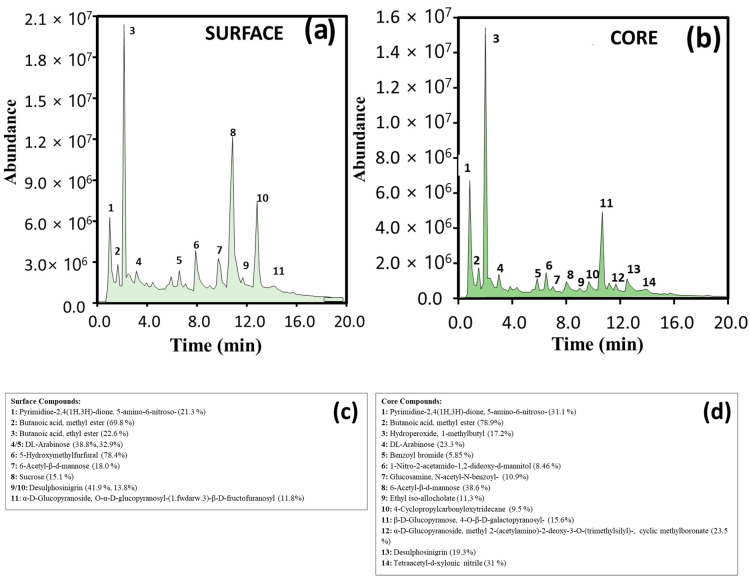
Total ion chromatograms (TICs) of GC-MS analysis of (**a**) kiwi surface and (**b**) kiwi core taken following SPME using a PANI@CNT/CNC SPME MN. Associated list of compounds found within (**c**) kiwi surface and (**d**) kiwi core deduced using the NIST library database.

**Figure 10 sensors-23-02317-f010:**
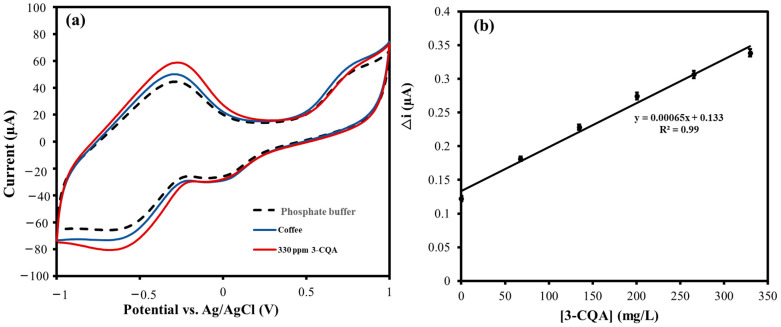
(**a**) Overlaid CVs obtained from the analysis of 0.10 M phosphate buffer blank, coffee (in 0.1 M phosphate buffer), and 330 mg/L 3-caffeoylquinic acid (3-CQA) (in 0.10 M phosphate buffer) taken using the PANI@CNT/CNC SPME MN. (**b**) Associated 3-CQA concentration vs. △Current standard addition plot.

**Table 1 sensors-23-02317-t001:** Electroactive surface area and electron transfer resistance (R_ct_) values for PANI@CNT/CNC SPME MN, CNT/CNC coated MN, and silylated MN.

Type	Electroactive Surface Area (cm^2^)	R_ct_ (kΩ)
PANI@CNT/CNC SPME MN	0.0215 ± 0.0005	1.06 ± 0.01
CNT/CNC coated MN	0.0070 ± 0.0004	6.0 ± 0.2
Silylated MN	-	10.0 ± 0.3

## Data Availability

All relevant data is included in the manuscript and in the Appendix A.

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
