# Peer review of "A Hybrid Stainless-Steel SPME Microneedle Electrode Sensor for Dual Electrochemical and GC-MS Analysis"

_sensors, 2023, doi:10.3390/s23042317_

Round 1

Reviewer 1 Report

Comment on the paper:

"A hybrid stainless-steel SPME microneedle electrode sensor for 2 dual electrochemical and GC-MS analysis” by M. Mugo et al.

The manuscript presents the research which used SPME coated microneedle for dual electrochemical and GS-MS analysis. The PANI@CNY/CNC SPME coated microneedle has increased the adsorption of analytes, thus increasing the sensitivity of the GS-MS analysis, and simultaneously can be used as in-situ electrochemical sensors to detect the analytes in fruit. The paper is well-written and easy to follow.

It is clear that the dual analysis has advantages, which combine the rapid detection using the electrochemical with the analytical analysis of the GS-MS technical. However, in the last experiment to detect 3-CQA, the author claims that the electrochemical method can qualify the substances. It would be more convincing if a reference measurement with other substances could be performed and presented in the paper.

Minor comments:

Some abbreviations should be explained in the paper.

The Y axis of figure 4. b should also be presented in the uA scale.

Author Response

Dear Editor                                                                                     

Title: Resubmission of manuscript ID: sensors-1917311 titled “A hybrid stainless-steel SPME microneedle electrode sensor for dual electrochemical and GC-MS analysis”

Please reconsider our attached revised manuscript for publication in the Sensors journal. The reviewers found our manuscript as interesting and of technical merit, while recommending some revisions. We have keenly revised and addressed all issues raised by reviewers.

As required, we append below in a point-by-point fashion how each of the reviewers’ comments was addressed. The responses are in blue.

Referee1
The manuscript presents the research which used SPME coated microneedle for dual electrochemical and GS-MS analysis. The PANI@CNY/CNC SPME coated microneedle has increased the adsorption of analytes, thus increasing the sensitivity of the GS-MS analysis, and simultaneously can be used as in-situ electrochemical sensors to detect the analytes in fruit. The paper is well-written and easy to follow. It is clear that the dual analysis has advantages, which combine the rapid detection using the electrochemical with the analytical analysis of the GS-MS technical. However, in the last experiment to detect 3-CQA, the author claims that the electrochemical method can qualify the substances. It would be more convincing if a reference measurement with other substances could be performed and presented in the paper.”

“Minor comments:

“Some abbreviations should be explained in the paper.”

We have gone through the manuscript and ensured all the abbreviations have been defined at their first place of usage.

The Y axis of figure 4. b should also be presented in the uA scale.

The figure has been corrected

We confirm our manuscript meet all the editorial office comments recommended.

Thank you for reconsidering our revised manuscript.

Prof. Samuel Mugo

Reviewer 2 Report

The authors have demonstrated the potential of a stainless-steel microneedle solid phase microextraction (SPME) platform for dual electrochemical and chromatographic detection for different applications

The experimental method is valid and was supported by arguments from the literature. There are new results and authors demonstrate how these can be used on different samples

So I recommend to accept after minor revision (clarifications to methodology)

The authors should compare this study to their own previous work [Mugo SM, Dhanjai, Lu W, Robertson S. A Multipurpose and Multilayered Microneedle Sensor for Redox Potential Monitoring in Diverse Food Analysis. Biosensors (Basel). 2022 Nov 10;12(11):1001. doi: 10.3390/bios12111001. PMID: 36354510; PMCID: PMC9688395] which was not mentioned in this paper.

Author Response

Dear Editor 

Title: Resubmission of manuscript ID: sensors-1917311 titled “A hybrid stainless-steel SPME microneedle electrode sensor for dual electrochemical and GC-MS analysis”

Please reconsider our attached revised manuscript for publication in the Sensors journal. The reviewers found our manuscript as interesting and of technical merit, while recommending some revisions. We have keenly revised and addressed all issues raised by reviewers.

As required, we append below in a point-by-point fashion how each of the reviewers’ comments was addressed. The responses are in blue.

Referee2

The authors have demonstrated the potential of a stainless-steel microneedle solid phase microextraction (SPME) platform for dual electrochemical and chromatographic detection for different applications

The experimental method is valid and was supported by arguments from the literature. There are new results and authors demonstrate how these can be used on different samples

So I recommend to accept after minor revision (clarifications to methodology)

As suggested by the reviewer, we have gone through the methodology and made substantial changes we thought could make the manuscripts more clear.

 The authors should compare this study to their own previous work [Mugo SM, Dhanjai, Lu W, Robertson S. A Multipurpose and Multilayered Microneedle Sensor for Redox Potential Monitoring in Diverse Food Analysis. Biosensors (Basel). 2022 Nov 10;12(11):1001. doi: 10.3390/bios12111001. PMID: 36354510; PMCID: PMC9688395] which was not mentioned in this paper.

As suggested by the reviewer, we have compared the characterization data of the sensor in this study with that published in the Biosensor article. The article has been cited.

 We confirm our manuscript meet all the editorial office comments recommended.

Thank you for reconsidering our revised manuscript.

Prof. Samuel Mugo

Reviewer 3 Report

Manuscript Number: Sensors-2184350

Title: A hybrid stainless-steel SPME microneedle electrode sensor for 2 dual electrochemical and GC-MS analysis 

The paper presents the easy and straightforward recognition of chiral redox active molecules based on the clockwise or anticlockwise rotation. Generally, the writing of paper is not qualified and informative and, therefore it is difficult to understand. The results aren't confirmed with sufficient information, and not all of it is well-described. There are several points which the author may like to consider in order further to improve the quality of the manuscript that has to be discussed as follows. Overall, the manuscript will be ready for publication after major revision.

1. Have the authors evaluated the storage stability of the electrode involved in the methodology?

2. What’s the affinity constant of the modified electrode and caffeine with the target?

3. In EIS. What is it: "applying a DC potential"? ..the oscillation potential? Range of frequency? What is the circuit model?

4. What is the irreplaceability of (PANI@CNT/CNC in this work? Could the authors give more discussion about this?

5. The selectivity of method has been evaluated.

6.  The mass spectrum of analyte and internal solution has been illustrated.

7. The variation of signals vs. concentration in electrochemical is lower in limit of noise.

8. The modification of the electrode is unclear. The authors have describe, regarding to the surface of needle has been modified, how the surface of needle not destroyed in during of injection port of GC.

9.  Please investigate the interday and intraday measurements and report with standard t-test results with confident level of 95% (alpha = 0.05), ensuring the robustness of the developed method.

10. All analytical performance provided in the figure need to be discussed more thoroughly. For instance, please discuss why the %recovery obtained is good enough (probably with standard t-test compare the spiked concentrations to the detectable concentrations) and why obtained %RSD provided good results.

Author Response

Dear Editor                                                                                                     February 5, 2023

Title: Resubmission of manuscript ID: sensors-1917311 titled “A hybrid stainless-steel SPME microneedle electrode sensor for dual electrochemical and GC-MS analysis”

Please reconsider our attached revised manuscript for publication in the Sensors journal. The reviewers found our manuscript as interesting and of technical merit, while recommending some revisions. We have keenly revised and addressed all issues raised by reviewers.

As required, we append below in a point-by-point fashion how each of the reviewers’ comments was addressed. The responses are in blue.

Referee3

“The paper presents the easy and straightforward recognition of chiral redox active molecules based on the clockwise or anticlockwise rotation. Generally, the writing of paper is not qualified and informative and, therefore it is difficult to understand. The results aren't confirmed with sufficient information, and not all of it is well-described. There are several points which the author may like to consider in order further to improve the quality of the manuscript that has to be discussed as follows. Overall, the manuscript will be ready for publication after major revision.”

We have thoroughly gone through the manuscript and made substantial changes by explaining clearly the significance of the work, clarified the methodology used and compared the results obtained in this study to those in other published work. We believe the quality of the revised manuscript is significantly enhanced for publication.

  1. Have the authors evaluated the storage stability of the electrode involved in the methodology?

We have clarified in the manuscript that PANI@CNT/CNC MN sensors are highly robust under storage conditions. This work was done over a period of one month with sensors showing no degradation or loss of performance over time. As shown in the methodology, even at high temperatures of 180 oC used for GC/MS analysis of the caffeine samples, the PANI@CNT/CNC MN sensors could be used many times (tested over 15 injections for three different needles) of a 400 mg/L caffeine standard over the one month period, with the needles kept in room temperature after use. Each needle was analyzed in triplicate during each injection cycle. The %RSD between the runs on same day on the same SPME sensor averaged around 1.6%, while the average across the 15 different runs for the three different sensors was 2.5%, which confirms the excellent repeatability and stability of the sensor under room temperature conditions. This has been clarified and included in the manuscript.

To confirm the reproducibility of the PANI@CNT/CNC MN sensors under electrochemical conditions, we further tested triplicate sensors for over one month period, analyzing a 300 mg/L 3-CQA for ~15 times with only ~1.7%% RSD distribution in the data distribution for each day, and ~3.0% RSD when data was averaged across all the runs for all the three sensor electrodes.

  1. What’s the affinity constant of the modified electrode and caffeine with the target?

The relative affinity of caffeine to the PANI@CNT/CNC SPME MN was compared with commercial SPME fibers  (based on DVB/CAR/PDMS ) by taking the ratio of the calibration sensitivity of the two platforms. As indicated in the manuscript, the PANI@CNT/CNC SPME MN was found to have 67% increase in sensitivity which can be attributed to the inherent high surface area impacted by the synergy of PANI@CNT/CNC hybrid materials for the SPME MN platform. This is consistent with literature where some conductive polymers have found to have excellent sorption for polar analytes. We have made that clear in the revised manuscript.   

  1. In EIS. What is it: "applying a DC potential"? ..the oscillation potential? Range of frequency? What is the circuit model?

The parameters used for setting up the EIS have clarified in the methodology section.

  1. What is the irreplaceability of (PANI@CNT/CNC in this work? Could the authors give more discussion about this?

I assume the reviewer means reproducibility of the PANI@CNT/CNC? The term ‘irreplaceability’  is rather confusing as it means impossible to replace. Nonetheless, in the revised version we have clarified the PANI@CNT/CNC is inexpensive platform as explained in the rather simple fabrication process. We have also clarified the  PANI@CNT/CNC yields highly reproducible results when used multiple times in both electrochemical and GC/MS conditions. This information has been included in the revised version.

  1. The selectivity of method has been evaluated.

It has been clarified in the manuscript that the  PANI@CNT/CNC MN sensor is analyzing electroactive compounds based on their reduction potential. As such, using cyclic voltammetry differentiation of diverse electroactive analytes would be based on the peak currents at different voltages. While this does not yield perfect selectivity as certain electroactive analytes may have overlapping redox peaks, it should be noted that highly selective sensors can be obtained using the PANI@CNT/CNC MN platform by integrating additional layers of biological and biomimetic molecular receptors. For example, integration of molecularly imprinted polymers has been demonstrated using similar platforms by imprinting neonicotinoids, and thus the platform can be highly selective and versatile with additional analyte specific molecular receptor.

  1. The mass spectrum of analyte and internal solution has been illustrated.

As suggested by reviewer, this has been included in Figure 6.

  1. The variation of signals vs. concentration in electrochemical is lower in limit of noise.

The question from the reviewer was not phrased clearly. So it was difficult to fully determine how to respond to this. I suspect however the extensive revisions we have included  in the revised version addresses sufficiently their comment.

  1. The modification of the electrode is unclear. The authors have describe, regarding to the surface of needle has been modified, how the surface of needle not destroyed in during of injection port of GC.

We have clarified the fabrication process of the triplicate  PANI@CNT/CNC MN platform. The PANI@CNT/CNC are entrained within the stainless steel needle and as such during injection, the polymer coating cannot be destroyed. This has been clarified in the manuscript. 

  1. Please investigate the interday and intraday measurements and report with standard t-test results with confident level of 95% (alpha = 0.05), ensuring the robustness of the developed method.

We have included the %RSD data for both the caffeine standard analyzed for 15 times over a period of one month using triplicate  PANI@CNT/CNC MN platform via GC/MS. Despite the high temperature of the GC/MS, the needles were stable and no degradation was observed as the average %RSD was 2.5%. This is not surprising as CNT/CNC and PANI highly stable under ambient and high temperature conditions. We have also included the % RSD for a similar analysis using electrochemistry of 3-CQA, which was also lower ~3% RSD for inter-sensor reproducibility, similarly indicating the robustness of these sensors.

  1. All analytical performance provided in the figure need to be discussed more thoroughly. For instance, please discuss why the %recovery obtained is good enough (probably with standard t-test compare the spiked concentrations to the detectable concentrations) and why obtained %RSD provided good results.

We have in detail discussed the analytical performance of the PANI@CNT/CNC MN by clearly explaining the calibration plots obtained, both external calibration and also standard addition calibration which caters to the matrix effects. We have also included the data on % RSD within one sensor and across triplicate sensors.

 We confirm our manuscript meet all the editorial office comments recommended.

Thank you for reconsidering our revised manuscript.

Prof. Samuel Mugo

Reviewer 4 Report

This is a very interesting article where the authors discuss a hybrid stainless-steel SPME microneedle electrode sensor that can be used for electrochemical and GC-MS analyses. Even though the manuscript is well written, overall, the paper still has significant room for improvement, and the authors are suggested to address the following points.

1) The authors have only cited one work from 2021 and one from 2022; this shows that they still need extensive background research. Please address this gap.

2) The conclusion section could be stronger. The authors have used the conclusion section to restate their results, making it less meaningful. Please ensure that the conclusion section discusses any unresolved questions/gaps / future directions of your works and their present implications. The conclusions section also needs to be connected to the original question and hypothesis that is being investigated.

3) Make sure that your materials section discusses where every chemical was sourced from, manufacturer details, etc., in detail so that anyone who attempts to reproduce your work can do so easily. In its present state, the manuscript needs more details in the material section. Please address this by providing the CAS number or product number in this section or in supporting information.

Author Response

Dear Editor                                                                                                     February 5, 2023

Title: Resubmission of manuscript ID: sensors-1917311 titled “A hybrid stainless-steel SPME microneedle electrode sensor for dual electrochemical and GC-MS analysis”

Please reconsider our attached revised manuscript for publication in the Sensors journal. The reviewers found our manuscript as interesting and of technical merit, while recommending some revisions. We have keenly revised and addressed all issues raised by reviewers.

As required, we append below in a point-by-point fashion how each of the reviewers’ comments was addressed. The responses are in blue.

Referee4

This is a very interesting article where the authors discuss a hybrid stainless-steel SPME microneedle electrode sensor that can be used for electrochemical and GC-MS analyses. Even though the manuscript is well written, overall, the paper still has significant room for improvement, and the authors are suggested to address the following points.

1) The authors have only cited one work from 2021 and one from 2022; this shows that they still need extensive background research. Please address this gap.

 The coupling of GC MS and Electrochemistry is generally unexplored. While we have added one additional 2022 article, there is generally limited literature on this, which demonstrates that significance of the work in this manuscript.

2) The conclusion section could be stronger. The authors have used the conclusion section to restate their results, making it less meaningful. Please ensure that the conclusion section discusses any unresolved questions/gaps / future directions of your works and their present implications. The conclusions section also needs to be connected to the original question and hypothesis that is being investigated.

 We have expanded the conclusion to make more informative and include future directions of the research work.

3) Make sure that your materials section discusses where every chemical was sourced from, manufacturer details, etc., in detail so that anyone who attempts to reproduce your work can do so easily. In its present state, the manuscript needs more details in the material section. Please address this by providing the CAS number or product number in this section or in supporting information.

 As suggested, we have clarified the grade of all the reagents used and where they were sourced in the materials and methods. It would be rather redundant information to include catalogue numbers.

 We confirm our manuscript meet all the editorial office comments recommended.

Thank you for reconsidering our revised manuscript.

Prof. Samuel Mugo

Round 2

Reviewer 3 Report

The authors tried their best to respond the reviewers' comments and questions, and to improve the quality of the manuscript. I believe the manuscript  could be accepted and published in Sensors.

Reviewer 4 Report

Satisfied with the author response.